# Tool and Workpiece Condition Classification Using Empirical Mode Decomposition (EMD) with Hilbert–Huang Transform (HHT) of Vibration Signals and Machine Learning Models

**Isaac Opeyemi Olalere *** and **Oludolapo Akanni Olanrewaju ***

Department of Industrial Engineering, Durban University of Technology, Durban 4000, South Africa
* Correspondence: Isaaco@dut.ac.za (I.O.O.); oludolapoo@dut.ac.za (O.A.O.)

**Abstract:** Existing studies have attempted to determine the tool chipping condition using the indirect method of data capture and intelligent analysis techniques considering machine parameters, and tool conditions using signal processing techniques. Due to the obstructive nature of the machining operation, however, it is daunting to use signal capturing to intelligently capture the condition of the tool as well as that of the workpiece. This study aimed to apply some advanced signal processing techniques to the vibration signals captured experimentally during machining operation for the decision making and analysis of tool and workpiece conditions. Vibration signals were captured during turning operations while using four (4) classes of tools, based on their flank wear. The signals were first pre-processed and decomposed using the Empirical Mode Decomposition (EMD) method. The Hilbert–Huang transform (HHT) was applied to the resulting IMFs obtained to compute the feature vectors used to classify the condition of the tool and workpiece. A total of 12 features, consisting of instantaneous properties such as instantaneous energy, instantaneous frequencies, and amplitudes, were obtained for data training and classification of tool conditions. To optimize the classification process, feature selection was performed using a genetic algorithm (GA) to reduce the number of features from 12 to 4 for data training and classification. The feature vectors were first trained for tool classification with a neural network scaled conjugate gradient (SCG) algorithm. The result showed that the model classification error was 0.102. Two other machine learning models, support vector machine (SVM) and K-Nearest Neighbors (KNN), were also implemented for classifying the tool conditions, from the feature vector, to determine the model that most accurately predicted the condition of the tool. To avoid bias and reduce misclassification errors, the k-fold cross-validation technique was applied with 'k' taken as 5 and 10. The computed feature vectors were used as inputs to train the machine learning model using both SVM and KNN models to classify the tool and workpiece condition during machining. The error loss of each model was evaluated and plotted to review the performance. The average overall error loss of 0.5031 was observed for the SVM model with 5-fold cross-validation, whereas the error loss of 0.0318 was observed for the KNN model with 5-fold cross-validation. The average overall error loss of 0.5009 was observed for the SVM model with 10-fold cross-validation when trained using the features selected by a genetic algorithm (GA), while the average overall error loss of 0.0343 was observed for the KNN model. The optimal performance of the SVM model was obtained when all features were used for the training, whereas the KNN model performed better when feature selection was implemented. The error losses of the models were evaluated to be less in KNN models, compared to SVM and SCG. The obtained results also showed that the developed KNN models performed 10 times better than the SVM model in predicting the tool condition from the captured vibration signal during the machining process.

**Keywords:** Machine Learning; Scaled Conjugate Gradient (SCG); Empirical Mode Decomposition (EMD); tool and workpiece condition; tool wear; Generic Algorithm (GA); Hilbert–Huang Transform (HHT); Support Vector Machine (SVM); Kernel Neural Network (KNN)

## 1. Introduction

Smart manufacturing has become the focus of most manufacturing companies due to product requirements, materials, technology, and data processing techniques [1]. Achieving the required product quality at a machining station is a daunting task due to the operating condition of the machine tool and the cutting tool condition. Tool wear/chipping directly impact the quality of the workpiece/product output from the machining station and the production cost. Chung, Wang [2] researched the effect of the complexity of product output on the cutting tool and the cost of machining and, hence, optimized some turning parameters concerning the tool conditions and the machining cost. Besides optimizing the turning parameters and machining cost, estimating the tool condition before and during the machining operation can also provide a knowledge base for determining the workpiece/product quality output. Tool wear/chip measurements have been evaluated using different direct and indirect techniques. Whereas the former method tries to measure the threshold of the wear/chip from the cutting tool, the latter estimates the wear/chip threshold from the effect caused by it [3–5]. The direct measurement approach evaluates the chip size of the tool using different techniques and tools that can measure the chip size. The shortfall of this approach is that it requires intermittent stopping of the machining operation for the measurement to be taken. However, this limitation was avoided in the study by Junaid, Siddiqi [6] using a computer vision-based system for measuring the dimension of an object in real-time during the machining process. On the other hand, the indirect method estimates the tool wear/chip condition by measuring the corresponding effect caused by the condition during machining [7]. Machine tool conditions during operation, such as the vibration of the tail-stock, the tool post, and the acoustics from the operation, may be evaluated to diagnose the condition of the tool and workpiece during operation. Furthermore, due to the obstructive nature of the machining operation, this method has been used by several research. In the current study, an integrated direct and indirect approach for tool condition monitoring was developed that is capable of non-obstructively evaluating the tool condition during machining.

Smart machining manufacturing is superseding conventional methods in precision manufacturing and optimization of facility outlay, resulting in improved productivity. The concern in machining operation is the tool condition and the quality of the product output. While the former affects the latter, the specifications of the product quality finish determine whether or not the product will be accepted, re-worked, or scrapped. Damage to the cutting tool during the machining process may cause either an upper-bound or lower-bound approach. The former affects only the cutting tool whereas, in the latter, damage to the cutting tool directly affects the workpiece, causing it to be scrapped. This directly increases the cost of production as both the tool and workpiece are replaced. Şap, Usca [8] investigated the effects of different machining parameters on surface roughness, tool wear, cutting temperature, and chip formation while turning Cu/Mo-SiCp composites. The result indicates that the most important parameter affecting surface roughness, tool wear, and the cutting temperature was the particle additive ratio. Usca, Uzun [9] also evaluated the tool wear, surface roughness, cutting temperature, and chip morphology using an Al/TiN-coated carbide cutting tool and found that optimization methods during the machining of particles play a significant role in extending the tool life considering the nature of the machining environment. Machining conditions and the machining environment significantly affect both the quality of the product output and the cutting tool wear at a machining station [10]. Many machining stations have resulted in a just-in-time maintenance policy that replaces the cutting tool at a specified time before the end of its useful life [11]. The shortfall of this approach is that the cutting tool is under-utilized, which also increases the running cost during machining. This has encouraged many research approaches to adopt smart manufacturing, because of the possibility of incorporating product quality requirments and manufacturing conditions into the manufacturing system, using fourth industrial revolution technologies such as smart sensors, smart IoT devices, cloud computing, and machine learning (ML) techniques [12]. Data processing and analysis for decision making

are essential for smart machining. Most approaches have adopted parameters from the tool, workpiece, and machine tool for decision making at the machining station. Several studies have shown that acoustic and vibration signals can be used for intelligent decision making using machine learning techniques. Although a vibration signal may accurately reflect the condition of the tool during operation, much effort is required to extract the significant component of the signal relating to the tool and workpiece condition during operation.

Cutting tool failure is a complex phenomenon that requires painstaking study, as the effect impacts the productivity, in diverse ways, of a machining station. Tool cracking/fracture (chipping) happens when a small fragment breaks from the cutting [13]. Although this condition may occur during the cutting operation, it may also happen during any phase of the tool's useful life. Tool wear, on the other hand, is the gradual change in the geometry of the cutting tool due to the progressive removal of materials [14]. Even though this condition happens gradually over a period during the useful life of the tool, it affects and determines the quality of the product/workpiece at a machining station. Another tool condition that affects the workpiece/product quality during the machining process is tool breakage. This happens when the cutting edge of the cutting tool breaks off [15].

Gradual or sudden deterioration of the cutting tool condition often results in the highlighted tool failures or conditions depending on the operating conditions. These failures are mainly due to two (2) mechanisms, which are the abrasion/friction between the interface of the tool and workpiece, and the adhesion due to the plastic deformation of the workpiece material [16]. This makes the unified study of both the workpiece and tool condition compelling. Khan and Gupta [17] carried out a study on the effect of the operating parameters on the cutting tool wear, considering cutting velocity, feed rate, and depth of cut, and found that tool wear was found to increase with increased feed rate and depth of cut. A similar result was shown by Roy [18], i.e., that cutting speed and depth of cut have significant impacts on principal and flank wear, with the latter lower than the standard limit of 0.2 mm with the maximum surface roughness of 0.99 μm. These studies showed that the operating conditions of the machine tool, such as the depth of cut, feed rate, and cutting speed, have a significant effect on the tool wear condition during machining.

A comparative study was also carried out on tool faults using vibration and cutting force signals to classify the conditions of the tool as healthy, worn flank, broken insert, and extended tool [19]. This study indicated that the conditions of the machine tool, such as vibration and acoustics, could influence or indicate the tool condition and be used as a factor for evaluating the tool condition. A similar study was carried out on the effect of vibration and cutting zone temperature on the surface roughness of workpiece and tool wear in an eco-friendly minimum quantity lubrication (MQL) at a constant cutting depth and feed rate. The results showed that, at a reduced vibration signal and temperature, there was also reduced tool wear and surface roughness [20]. Another study carried out by Rizal, et al. [21] classified different levels of tool wear in the milling process, using vibrations, tool tip temperature, and cutting force, into medium and critical wear stages. This literature has illustrated that there is a link between the tool condition, the surface roughness of the workpiece, and the machine tool operating parameters during machining. However, extracting more features from the captured vibration signal may provide better analytical results when trained using machine learning (ML) algorithms. A larger number of extracted features provides more available data for the classification algorithm.

Presently, most manufacturers use time as a conventional tool replacement strategy subject to the operator's experience [22]. This strategy often results in either an early replacement (under-utilizing the tool) or a late replacement, which often causes damage to both the workpiece and machine tool. Therefore, tool condition monitoring (TCM) has been considered a key enabling technology for manufacturing optimization [23]. The system estimates the tool condition by either deploying sensor-based models or analytical models [24].

Plaza, López [25] revealed that not all vibration feature extraction methods are appropriate for real-time monitoring of the surface finish, but the vibration signal with the

wavelet packet transform (WPT) method can effectively be used for real-time surface finish monitoring, with high accuracy and reliability, and a low computational cost, in CNC machining. The study shows that the vibration signal is a better feature than the acoustic emission (AE) in monitoring the surface finish of the workpiece. In addition, Kiew, Brahmananda [26] showed that tool wear and the machine vibration signal are related to each other at varying depths of cut and feed rates in different experiments. Although Ochoa, Quinde [27] proposed that the AE signal, when mounted on the tool holder rather than the workpiece, could enhance the reliability of the TCM system, in contrast, Deja and Licow [28] disproved the ability of AE sensors to distinguish between a new and worn-out tool when mounted on either the tool holder or the workpiece. It may therefore be of interest to note that the vibration signal is an important parameter in the TWCM system for depicting the state of the tool and the workpiece during the turning operation.

Sensor-based tool and workpiece condition monitoring systems comprise highly sensitive sensors integrated with IoT controllers, which help with signal interpretation, analysis, and decision making. Like TCM, these systems can be divided into broad methods, which are direct and indirect [29]. The direct TCM method directly measures the changes in the geometry of the cutting tool during the cutting process, whereas the indirect TCM method measures the online operating parameters as a way of detecting the tool's deteriorating condition. Due to the nature of the cutting region during the process, however, the direct TCM method is challenging. Because direct TCM methods are grueling to implement online, most studies have adopted indirect TCM methods [30]. Similarly, TCM methods can be extended to monitor the surface quality of the workpiece using the same approach. This implies that the accuracy and precision of the measured conditions of the cutting tool, and of the workpiece/product quality, directly inform the accuracy and precision of the indirect online TCM system based on the acquired signals and the robustness of the processing techniques. The measured parameters are the data that are interpreted by the monitoring system to evaluate the current condition and the changing conditions; hence, more detailed attention needs to be paid to this system.

Accelerometer sensors are devices that sense the displacement of a component or machine from its mean position [31]. The vibration effect during a cutting operation may result from several factors. A cutting tool that is broken or worn-out during a cutting operation can cause a resultant vibration effect on the machine tool. Similarly, a faulty jaw can induce spindle vibration, which affects both the tool condition and the workpiece [32]. Cheng and Dang [33] detected varying conditions, such as a normal spindle, an unbalanced rotor, a gearbox crack, and a bearing crack, on a CNC machine tool by analyzing vibration signals captured by accelerometer sensors installed on the spindle surface near the bearing. This captured the vibration signal resulting from malfunctioning of the machine tool component, which in turn affected the surface roughness of the workpiece and tool wear. In contrast, Munawar, Mufti [34] deployed a magnetic-type accelerometer, attached to the spindle bearing housing, to capture the vibration signals from the machine tool during the machining of AISI 1040 carbon steel.

Furthermore, Gao, He [35] noted that an abrupt increase in the spindle rotation frequency to the natural frequency of the spindle structure increased spindle vibration, which degraded both the surface of the workpiece and the spindle performance. Therefore, capturing the vibration signals using an accelerometer is of great interest in monitoring the surface roughness of the workpiece and the tool condition. Signals captured from the monitoring devices installed on the machine tool can be classified as steady-state signals and dynamic/transient signals. The former are captured when the condition of the machine is stable during operation, whereas the latter are captured during the unstable operating condition of the machine tool. These two distinctive categories of signals captured from the machine tool have been analyzed using different techniques and methods to extract intelligent information that is useful in depicting the state of the process. Different signal processing methods can be adopted for the TWCM system. Whereas time domain analysis techniques are used to evaluate physical signals and mathematical functions with reference

to time [36–39], frequency domain techniques are used to analyze signals or mathematical functions with reference to frequency [40–42].

Signals can be converted from either the time domain to the frequency domain or vice versa with an operator called a transform. Early research on the TCM system adopted the Fourier transform, which converts a time function into an integral of sine waves of various frequencies; however, this is deficient for the analysis of non-periodic and non-stationary signals, hence, other types of transforms have been developed [43]. Wavelet packet transforms (WPTs) have shown better results as a computational method for time–frequency signal conversion and, as a result, have been widely used in many tool condition monitoring studies [44–47]. A comparative study to predict bearing degradation [48] using a discrete wavelet transform (DWT), tabular Generative Adversarial Networks (TGANs), and ML models also showed that the DWT is an efficient signal processing tool for decomposing signals that are non-stationary signals. Other transforms exist for signal decomposition and computing time–frequency conversion, such as the Hilbert–Huang transform [49]. In addition, artificial intelligence techniques are another type of signal-processing method used for TCM systems [50]. This technique was adopted by Bhavsar and Vakharia [51] for fault prognosis and condition monitoring using the previous data to predict the remaining useful life (RUL) of the component using various regression/degradation models. Due to the robustness and strength of this technique in the analysis of large volumes of data, and the development of intelligent models for predicting the condition of the workpiece and tool, the usage of this method has increased recently in TCM systems [52,53]. Several AI algorithms have been deployed in either predicting or classifying tool or workpiece conditions at a machining station. The support vector machine (SVM) algorithm has been applied in many TCM systems for classifying tool conditions [54,55], and the artificial neural network (ANN) algorithm has been widely used for predicting tool conditions and workpieces [56]. A recently evolved deep learning AI algorithm finding its way into TCM systems is the convolutional neural network (CNN) [57]. The CNN algorithm takes in input data in the form of an image, processes it by extracting its features, and evaluates the cutting tool's condition. Ambadekar and Choudhari [58] developed a tool wear prediction system to monitor the flank wear of a cutting tool using a CNN, and concluded that the method gives a good response to the data in the form of images, with accuracy of 87.26%.

Several studies have been carried out on the evaluation of tool conditions and workpiece roughness. In this study, tool conditions were classified based on flank wear. The corresponding quality output of each tool class was experimentally determined by measuring the surface roughness parameters of the workpiece. The vibration signals were captured during machining operations using IoT-enabled sensors and gateways. The signals were decomposed by applying the EMD method to the raw vibration signals. This method was applied because of the nature of the vibration signals captured from the machining process. The Hilbert transform (HHT) was applied to the resulting IMFs from the decomposition to extract the feature vectors that were used to classify the cutting tool. To optimize the classification algorithm and make it less expensive, a genetic algorithm (GA) was used for feature selection. Finally, the features selected using the GA model after applying HHT to the decomposed signals were fed into machine learning (ML) models to classify each tool condition and the corresponding quality output. The classification was performed using three different ML models, and the error loss of each model was evaluated. Neural networks with SCG, KNN, and SVM algorithms were used to develop the classification models. The error loss of each model was evaluated to determine the optimal algorithm for the classification problem. The study can provide an alternative solution to intermittently stopping the machining process to evaluate the tool and workpiece condition during machining.

## 2. Materials and Method

### 2.1. Vibration Signal Capturing and Processing for TWCM System

The vibration signal during the machining operation was captured by installing an advanced vibration sensor, MNS 2-9-W2-AC-ADV, on the tool post of the machine tool. An Alta wireless vibration sensor enabled with IoT technology was used in this study, as indicated in Figure 1. The heartbeat of the sensor was adjusted to be captured per second. This enabled the measurement of the vibration signals corresponding to the varying tool and workpiece conditions. Abrupt or gradual changes in the condition of the tool and the workpiece could hence be continuously monitored by monitoring the vibration signal emanating from the machining station.

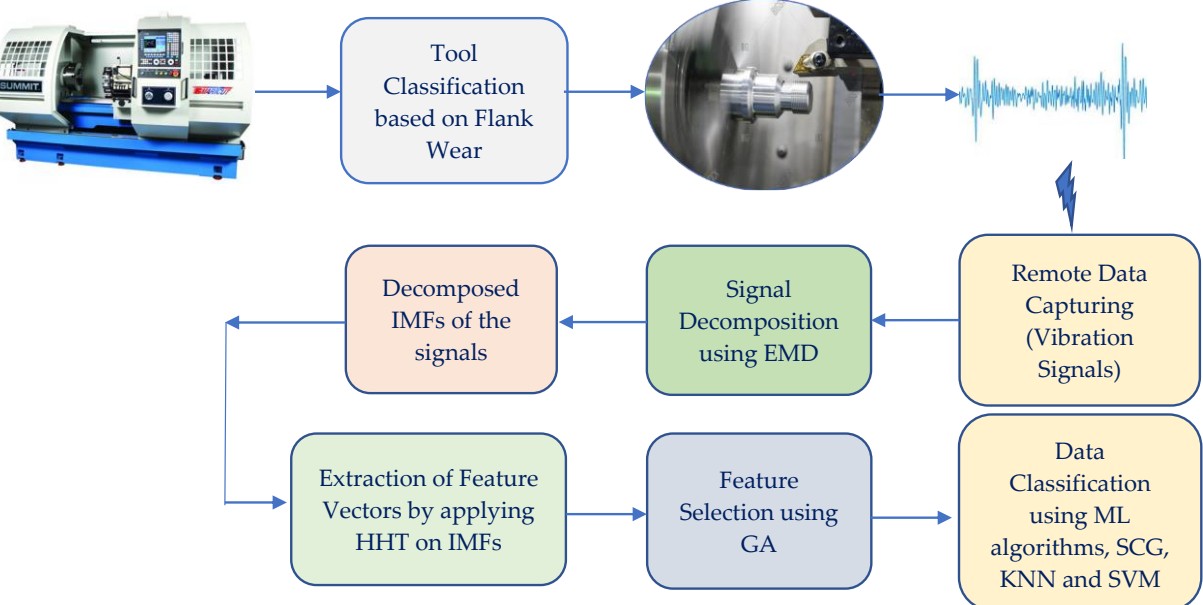

**Figure 1.** Flow Chart of the Proposed Methodology.

2.1.1. Experimental Set-Up

To determine the condition of the tool and workpiece during the operation based on the vibration signals, four (4) cutting tool categories were used for the machining experiment. An indexable tungsten cutting tool with CCMT09T3034 carbide inserts was used for the turning operations. The first class of tools was brand-new tools, the second class was used good tools, and the third and fourth classes were rough and worn-out tools, respectively. A new tool has 100% of its useful life remaining, while a worn tool has zero (0%) life remaining. A good tool has a useful life of about 70% or more. A worn tool has less than 50% useful life, however, it can be used depending on the surface finish requirement of the job or product. A standard recommended value in defining a tool life criterion based on ISO 3685:1993 requires that cutting tool inserts with flank wear of 0.3 mm be discarded. Rizal, Ghani [21] states that the ranges of flank wear values were divided into three classifications, normal wear (VB = 0–0.15 mm), medium wear (VB = 0.15–0.25 mm), and critical wear (VB = 0.25–0.35 mm), adhering to the ISO 3685:1993 standard. Similarly, this study classified tools using flank wear, where a new tool was presumed to have 0 flank wear, a good tool was taken to have the same range as that of normal wear (VB = 0–0.15 mm), a rough tool had medium wear (VB = 0.16–0.29 mm), and a worn tool had flank wear similar to the critical wear range (VB > 3.0 mm). As tools gradually move from one phase of useful life to another during the turning operation, it becomes very important to monitor the tool conditions, so as to optimize production by reducing the chances of nonconformities in the product quality. Bright carbon cylindrical steel workpiece materials, BS 970 080m40 and BS 970 070m55, were used for running the test, and the corresponding vibration during the

machining operation was recorded. The set-up of the back-end for remote collection of the captured data and installation of sensors for data capturing is illustrated in Figure 2.

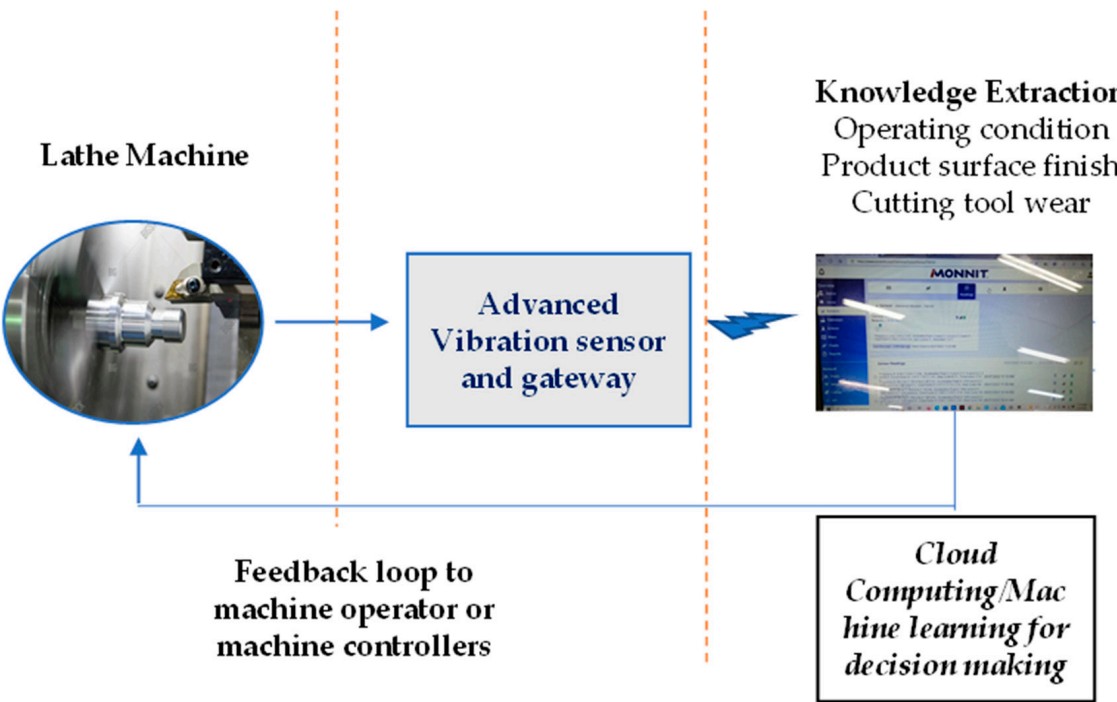

**Figure 2.** Image Showing the Server for Vibration Signal Capturing.

The experiment was repeated by capturing the vibration signals from the respective cutting tool class used for the machining process. The data were recorded as the captured vibration signals, the time intervals and labels that indicate the tool category used for the machining process. The vibration data may not be applied to analyses for decision making because they are time-series data, and because of the transient conditions that may have contaminated the precision of the signal. Hence, the data were processed using advanced signal analysis techniques, and a feed-forward neural network algorithm was employed to classify the signals into each condition class.

2.1.2. Advanced Vibration Signal Processing

Due to the nature of the machining operation, and the obstructive nature of the operation, the vibration signal needs to be processed for the proposed machine learning algorithm to achieve a great result. A graphical plot of the vibration data from the tool classes against the time interval is presented in Figure 3, for which the cutting tools were replaced at different intervals to capture the varying vibration signals being transmitted by different cutting tool classes. The plot illustrates the different vibration signals of each class of cutting tool used for the machining operation. The vibration signal can be observed on the graph. The excitation is firstly due to the alignment of the accelerometer (vibration sensor) to the gravitational field and, secondly, because of the power generated due to the reaction between the cutting tool and the workpiece during the turning operation. The vibration signals may vary as the tool gradually moves from one phase of the condition to another; therefore, to be able to discriminate the signals propagated as the tool condition changes, a new signal that can be differentiated based on the shape of oscillation and how quickly the signal varies over time was created from the vibration signals.

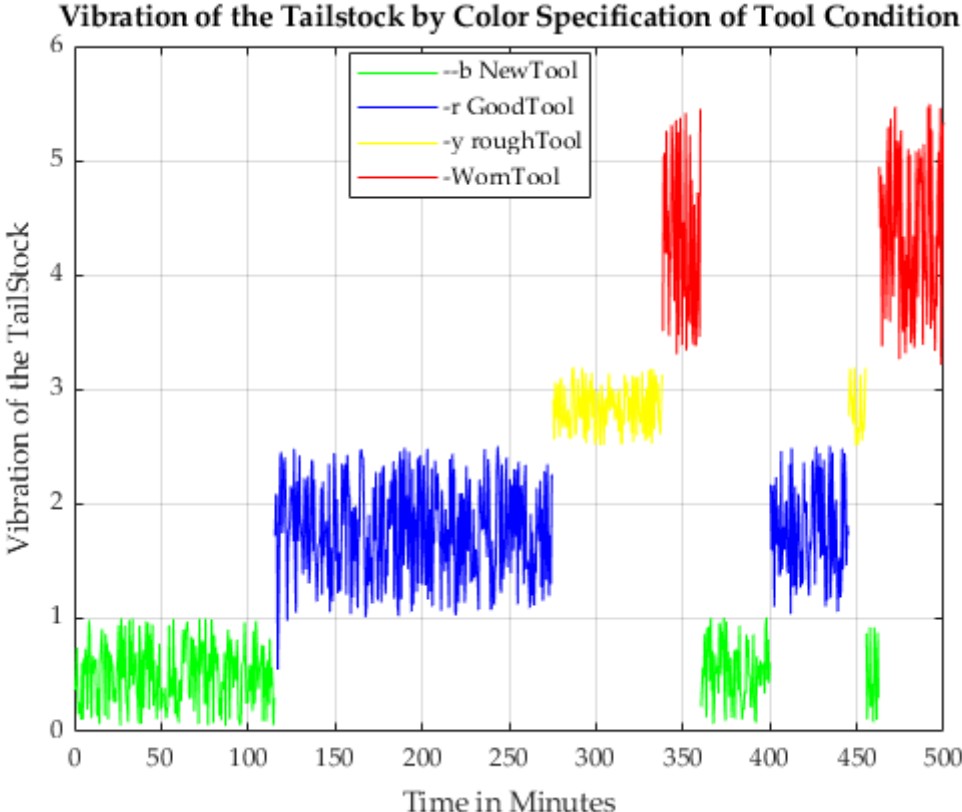

**Figure 3.** Graph of Vibration by Color Against Time.

The major characteristics of the vibration signals captured during the machining operation are the non-linearity and non-stationary natures of the signals. Non-stationary continuous signals are composed of sinusoidal waves with a distinct change in frequency. Therefore, a signal processing method capable of analyzing these signal characteristics must be adopted. Since the fast Fourier transform (FFT) is ideal for linear and stationary signals due to its uniform trigonometric function, wavelet transform was introduced as an alternative to extracting time–frequency resolutions of a signal. However, the EMD method is a better approach for analyzing non-linear and non-stationary signals. This is because the EMD method is based on the data, in contrast to most other methods. The EMD method decomposes signals into components to provide insight into inherent features. The signal is decomposed into a finite number of IMFs (real part) and the residual (imaginary part), as indicated in Equation (1):

$$f(t) = \sum_i imf_i + res \tag{1}$$

where $imf_f$ represents the intrinsic mode functions and the residual. The EMD function evaluates the local extrema of the signal and fits the maxima ($E_{up}(t)$) and the minima ($E_{low}(t)$) to an individual envelope. The mean of the upper and lower envelope is determined as indicated in Equation (2):

$$E_{mean}(t) = \frac{(E_{up}(t) + E_{low}(t))}{2} \tag{2}$$

The residual component of the signal is determined as indicated in Equation (3):

$$res(t) = f(t) - E_{mean}(t) \tag{3}$$

Since the process is finite, the stopping criterion is determined by Equation (4):

$$\sum t = \frac{(res(t) - f(t))^2}{f(t)^2} < \in \tag{4}$$

Therefore, the decomposition stops when the residual approaches a monotonic function. The signal is decomposed into IMFs and residuals, as indicated in Figure 4.

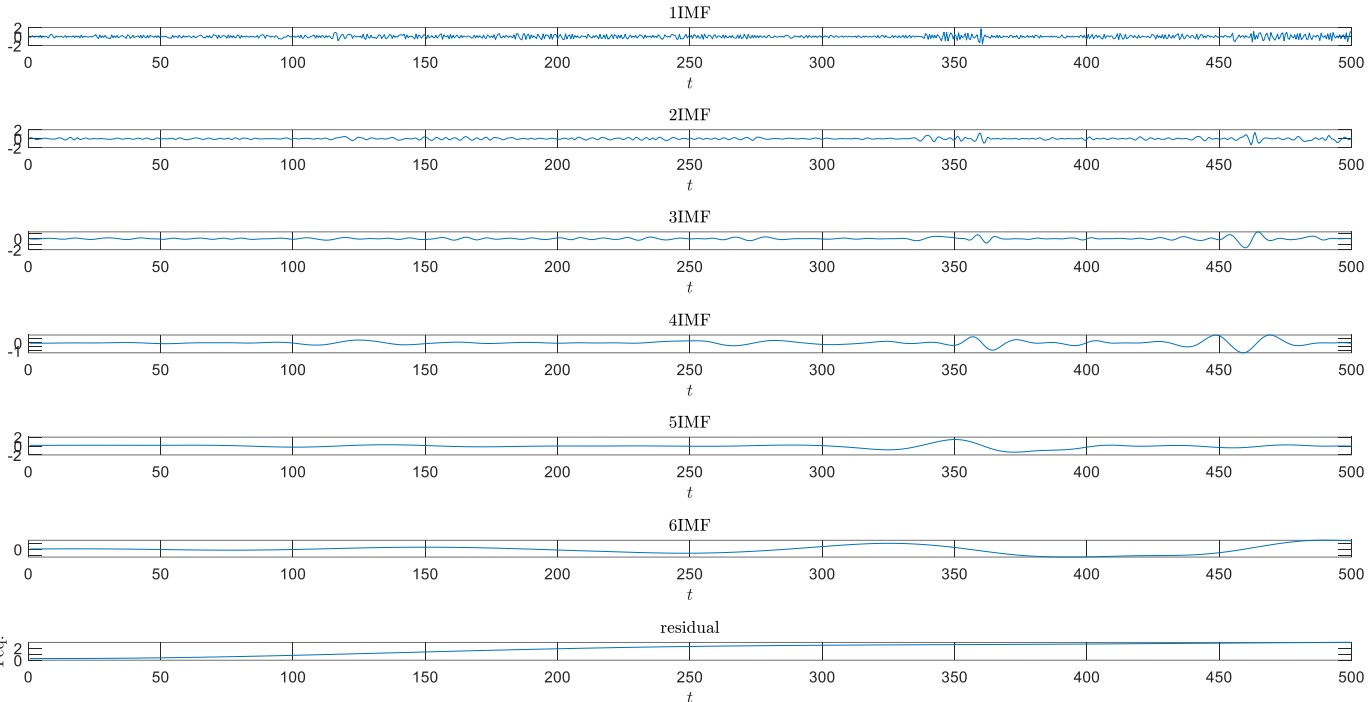

**Figure 4.** Decomposed Signals into IMFs and Residual.

To determine the instantaneous properties from the decomposed IMFs and residuals, the Hilbert–Huang transform (HHT) was applied. HHT was applied to compute the instantaneous energy and instantaneous frequency of each IMF mode. For each IMF, $x_i$, the HHT function computes the components as indicated in Equation (5):

$$x_i = f(i) + iH\{f(t)\} = A(t)e^{i\varphi(t)} \tag{5}$$

where $H\{x_i\}$ is the Hilbert transform of $x_i$, $A(t)$ is amplitude, and $\varphi$ is instantaneous phase. The amplitude and phase are expressed in Equations (6) and (7), respectively:

$$A(t) = \sqrt{f^2(t) + H\{f(t)\}^2} \tag{6}$$

$$\varphi(t) = arctan\left(\frac{H\{f(t)\}^2}{f(t)}\right) \tag{7}$$

The Hilbert transform provides a unique imaginary component, $H\{f(t)\}^2$, with the instantaneous energy given in Equation (8) and the instantaneous frequency in Equation (9):

$$\rho = |A(t)|^2 \tag{8}$$

$$\omega(t) = \frac{d\varphi(t)}{dt} \tag{9}$$

Since most frequencies are not continuous, the Hilbert transform is based on the discrete Fourier transform.

The feature vectors for classifying the conditions of the tools were computed by applying the Hilbert transform to the IMFs of the decomposed signals. The features were therefore the instantaneous frequencies, energy, and amplitude of the IMFs, as indicated in Equations (6)–(9), respectively. Since the instantaneous energy is very dependent on the amplitude, the feature vectors are usually the instantaneous frequency and energy for classification models. In this study, a total of 12 features were computed from the IMFs of the decomposed signals.

To reduce the computational cost of the classification model, hyperparameter optimization using the GA model was applied to the features to select the features that were essential for the learning algorithm. The number of features was reduced to four (4) after feature selection was undertaken using the genetic algorithm model, using the Roulette Wheel (RW) method. The fitness probability of a single chromosome in the generation was determined by Equation (10):

$$Fp = \frac{F_i}{\sum_{i=1}^{n} F_i} \tag{10}$$

where $Fp$, is the fitness probability of the $ith$ chromosome, and $F_i$ is the fitness value of the ith chromosome.

The feature vector was trained for data classification using the ML classification algorithm. The neural network SCG algorithm was first applied to train the data for classification. Furthermore, SVM and KNN learning models were also used for the classification model and the best-performing model was essentially determined from the loss function of each model.

## 3. Results and Discussions

Extracting features using the signal decomposition method with the Hilbert transform helps to provide detailed and useful information on each signal buffer produced by different classes of tools and workpiece conditions, and thus to optimize manufacturing. This implies that the condition of the tool and workpiece can accurately be determined during production while avoiding machine downtime, meeting job requirements, and reducing scraps due to product damage (i.e., surface quality exceeding the required standards). Due to the non-linear and non-stationary characteristics of the signal captured at the machining station, it is advantageous to use the signal decomposition function to identify the varying condition of the cutting tool during the machining operation. In a review, Bokde, Feijóo [59] showed that 80% of the most recent analyses of non-stationary and non-linear wind turbine signals adopted EMD for signal analysis for the prediction models. Compared to other decomposition methods, EMD generates relatively stationary subseries (IMFs) that are easily modeled [60]. Furthermore, IMFs developed with EMD eliminate stochastic volatility and, therefore, improve the prediction results [61]. The vibration signals captured during the machining operation were decomposed using the EMD method to extract the IMFs, as indicated in Equation (11):

$$[IMFs, \; res] = emd(data); \tag{11}$$

Six IMFs and one residual were obtained after the decomposition of the signals, as indicated in Figure 4. The iterative decomposition was stopped using the stopping criterion indicated in Equation (4).

The instantaneous properties of the signals were obtained by applying the Hilbert transform (HHT) to the IMFs. The obtained feature vectors comprised 12 features. These included the instantaneous frequencies, amplitude, and energy of the IMFs. These features were fed into the ML algorithm to classify the varying signals from each tool type. The feature vectors were first trained using a feed-forward neural network model with a scaled conjugate gradient (SCG) algorithm. The model had four (4) features, while the hidden layer had eighteen (18) nodes, and four (4) output classes were used. The label for the data

to be trained had one (1) column and a thousand (1000) samples. The network trained the data set using 44 iterations and the confusion matrix was used to evaluate the performance of the network. The best validation performance, as shown in Figure 5, was 0.047349 at epoch 103.

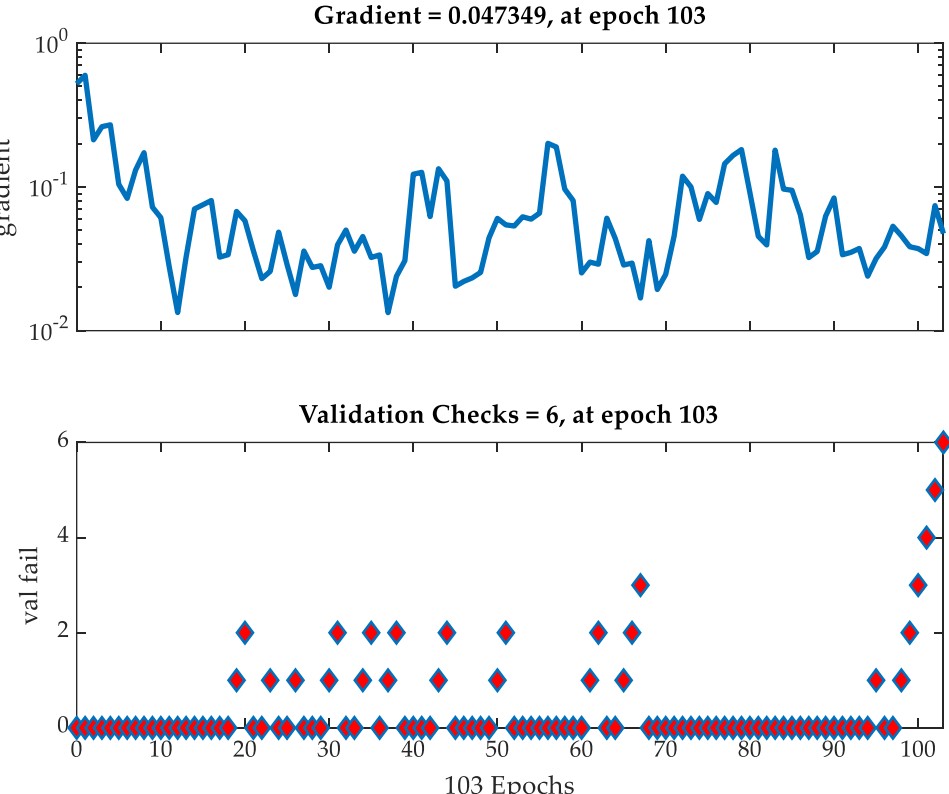

**Figure 5.** Training State of Neural Network with SCG algorithm.

The histogram of the error of the neural network with the SCG algorithm indicates the classification prediction error after training the feed-forward backprop neural network. It shows how the prediction class differs from the target class in a training example. This is very important in determining the accuracy of the trained network in classifying new samples of data. The error histogram has 20 bins, which indicate that the range of error bars on the histogram is divided into 20 samples, as shown in Figure 6. This is evaluated from Equation (12):

$$Bin\ Width = \frac{Right\ limit - lower\ limit}{20} \tag{12}$$

The bin width from Figure 6 can be evaluated to be 0.08882. Although the classification algorithm performs very well according to the error histogram, there are other error distributions to the left and right of the zero-error line.

The confusion matrix in Figure 7 shows that the accuracy of the classification algorithm is 89.2%.

To further determine the accuracy of the classification algorithm, other ML classification algorithms were considered for classifying the conditions of the cutting tool during machining operation from extracted vibration signals. The SVM is a powerful technique used in data classification and regression analysis, and it has become one of the most used classification methods due to its good theoretical foundations and good generalization capacity [62]. Kaya, Kuncan [63] applied SVM and logistic regression (LR) to classify the vibration signals at varying bearing speeds, and the result showed that the LR model yielded a poorer prediction. Similarly, whereas Chen, Li [42] classified vibration signals based on variational mode decomposition (VMD) and energy entropy using the SVM technique, for

fault diagnosis in rotating bearings, Glowacz, Glowacz [64] applied a linear support vector machine (LSVM) for the classification of data between two classes of signals captured from the vibration of an induction motor machine by finding the best hyperplane.

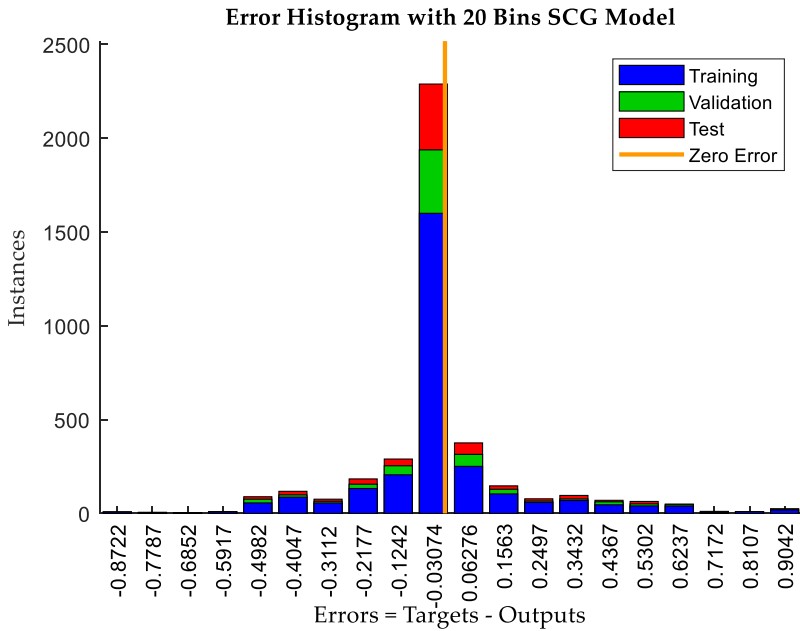

**Figure 6.** Error Histogram of SCG Neural Network Model.

**All Confusion Matrix**

|   | 1 | 2 | 3 | 4 | |
|---|---|---|---|---|---|
| **1** | **296** 29.6% | **31** 3.1% | **10** 1.0% | **2** 0.2% | 87.3% 12.7% |
| **2** | **11** 1.1% | **379** 37.9% | **1** 0.1% | **0** 0.0% | 96.9% 3.1% |
| **3** | **0** 0.0% | **0** 0.0% | **125** 12.5% | **19** 1.9% | 86.8% 13.2% |
| **4** | **18** 1.8% | **0** 0.0% | **10** 1.0% | **98** 9.8% | 77.8% 22.2% |
|   | 91.1% 8.9% | 92.4% 7.6% | 85.6% 14.4% | 82.4% 17.6% | **89.8%** **10.2%** |

Output Class / Target Class

**Figure 7.** All Confusion Matrix for SCG Neural Network Model.

However, Altaf, Akram [65] classified EMD features and FFT features extracted from vibration signals to diagnose bearing faults without any statistical information using KNN classifiers, with the method yielding a reduction percentage of 96.64%. The result of this classification algorithm showed a good performance, even though the study applied FFT to the decomposed signal, in contrast to this study, which applied HHT to the decomposed signal. Similarly, [66] applied both KNN and SVM to features extracted from vibration signals processed through FFT and principal component analysis (PCA), and showed that vibration signals are sufficiently rich in information about the machine to enable precision machining and 100% state classification accuracy to be achieved. Hence, SVM and KNN

were applied to the extracted features for the classification of the conditions in other phases to determine which classification model performs optimally for classifying tool classes.

Bias and misclassification error are mostly a challenge when applying ML algorithms and techniques to classification problems. Therefore, to avoid these, k-fold cross-validation techniques are applied. This study applied 5-fold and 10-fold cross-validation, and the models were compared to determine the one with the best performance in terms of the error loss in classification. For the 5-fold cross-validation technique, the models were developed and tested by determining the error loss for both SVM and KNN to determine the model with better performance. Figure 8 illustrates the error loss of each model of the SVM and KNN methods, with the blue color representing SVM models and red representing KNN models.

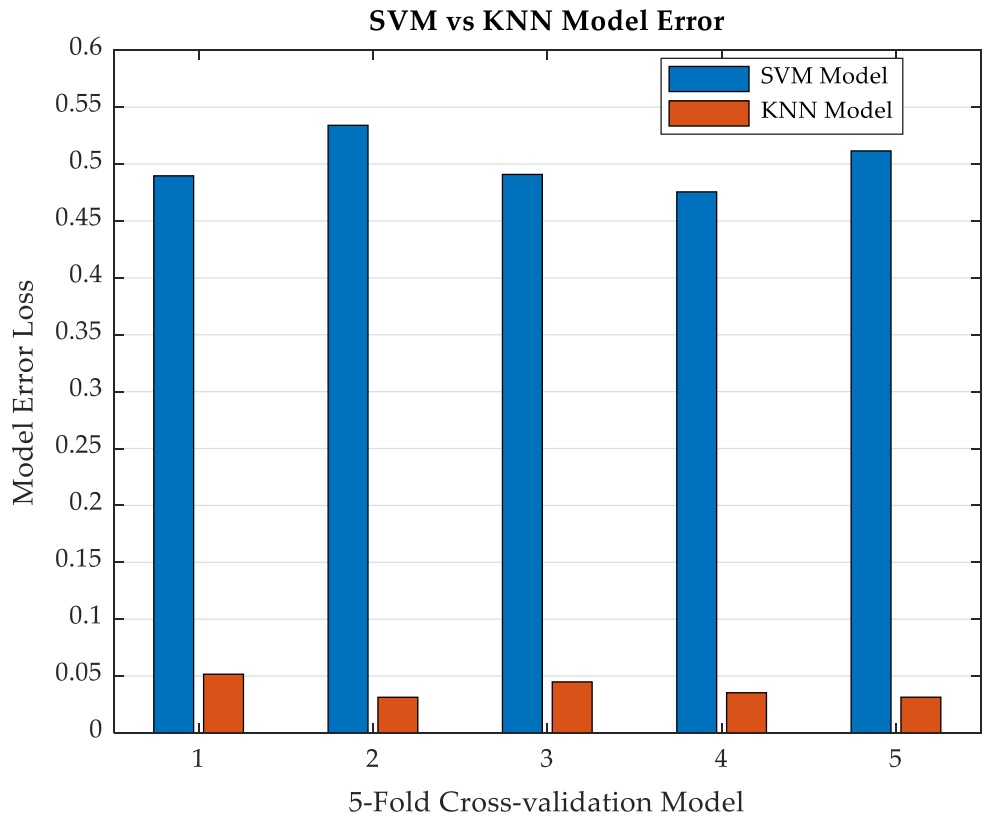

**Figure 8.** Error Loss for SVM vs. KNN classification with 5-fold Cross-validation.

The result showed that the KNN algorithm performs better than SVM in classifying the cutting tool classes during the machining operation. To determine the overall error loss for each of the models, Equation (13) was applied:

$$E = \frac{1}{k} \sum_{i=1}^{k} E_i \tag{13}$$

where *k* represents the number of folds being considered and *E* is the error loss. Therefore, the overall error loss for the five SVM models was 0.5031, while for the KNN model it was 0.0318. This indicates that KNN models performed better than the SVM models for tool condition classification during the machining process.

For the 10-fold cross-validation technique, both models were also developed to determine their performance. Figure 9 shows that the KNN models all performed better than the SVM models. Each SVM model from 1 to 10 had a higher error loss function compared to the KNN models.

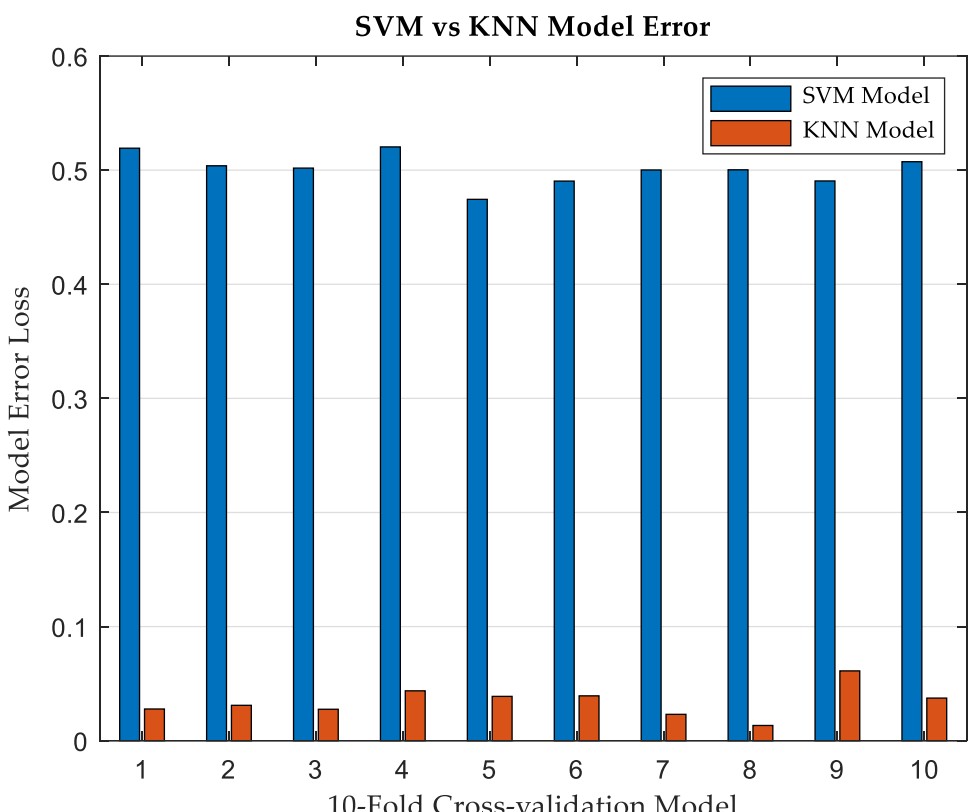

**Figure 9.** Error Loss for SVM vs. KNN classification with 10-fold Cross-validation.

The overall error loss for the SVM models with 10-fold classification was 0.5009, whereas the error loss for KNN models was 0.343. The general performance of the two (2) models shows that the error loss of SVM slightly improved when the 10-fold cross-validation technique was used, compared to the 5-fold cross-validation technique. Furthermore, the performance of KNN models was better with the 5-fold cross-validation technique than with the 10-fold cross-validation technique. In Figure 9, the KNN8 model is the best, with the least error loss, while the SVM5 model is best for the 10-fold cross-validation. Toledo-Pérez, Rodríguez-Reséndiz [67] reviewed the SVM-based model of EMG signal classification and reported that a large number of sounds, vibration signals, and images have been classified using the SVM classification algorithm, achieving more accuracy without feature selection and 5% less accuracy with feature selection. Therefore, to determine if the SVM model performs better without feature selection, the models were evaluated with the 12 features, and the loss function was determined. The performance of both models for 5-fold cross-validation with all the feature vectors is illustrated in Figure 10, which shows that, for SVM models, the performance improved greatly compared to when feature selection was implemented. The overall average error loss when 5-fold cross-validation was performed on all the features was 0.1668, compared to 0.5031 when feature selection was performed. However, for KNN models using 5-fold cross-validation with feature selection, the overall average error loss increased from 0.0318 to 0.2202. These results show that, although feature selection improves the performance of KNN models in classifying the conditions of the tool, this is not the case with the accuracy and performance of the SVM model.

The result shown in Figure 11, for the 10-fold error loss for SVM and KNN classification models developed without applying feature selection, also indicates that the performance of SVM models was more accurate without feature selection. The error loss for 10-fold cross-validation for SVM models when feature selection was applied was 0.5009, while it reduced to 0.1578 without feature selection. On the other hand, the performance of KNN models when feature selection was adopted was 0.0343, while it increased to 0.2172

without feature selection. Therefore, the KNN algorithm performed better in classifying the condition of the cutting tool during the machining operation.

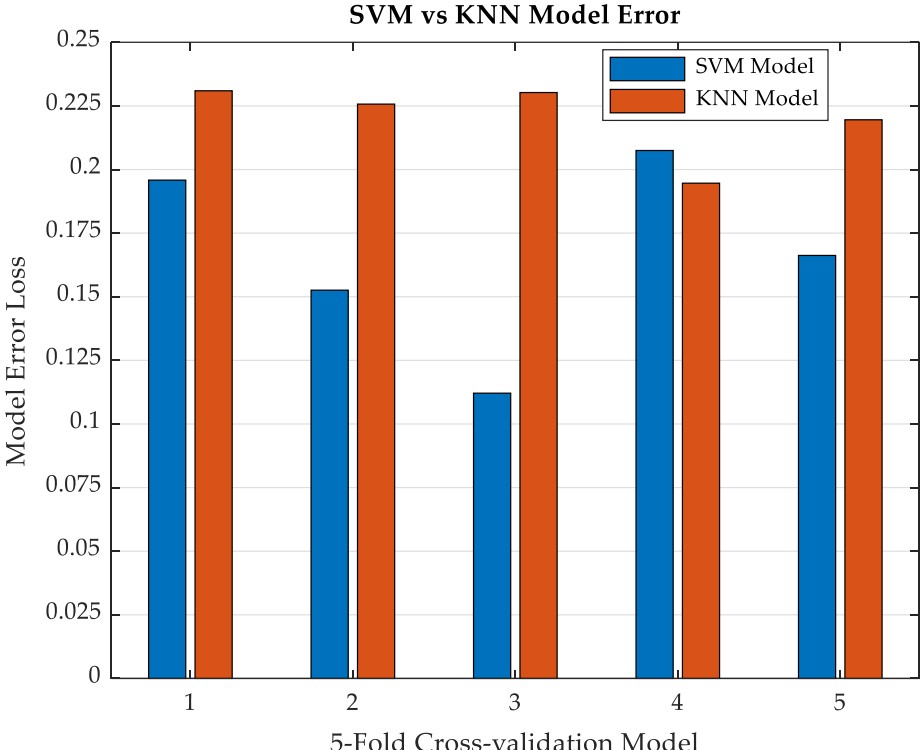

**Figure 10.** Error Loss for SVM vs. KNN classification with 5-fold Cross-validation without feature selection.

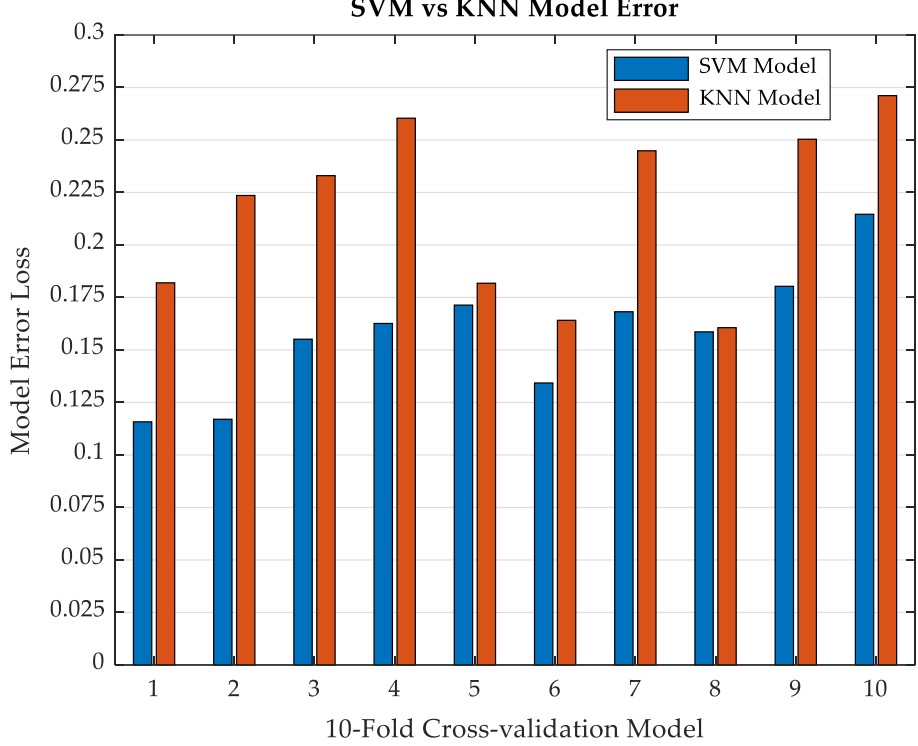

**Figure 11.** Error Loss for SVM vs. KNN classification with 10-fold cross-validation without feature selection.

## 4. Conclusions

Diverse approaches have been applied to initiating different measurable conditions on the cutting tool to generate signals during operation that can be analyzed to deduce the tool condition. Whereas some studies have artificially created a crack on the tool to capture signals (vibration, acoustics, etc.) capable of being analyzed to give feedback on the condition of the tool, some studies have measured the tool chip size to capture the signals produced by the tool condition and the product quality output. The former approach is limited by the possible tool damage or weakness caused by the method, whereas the latter is limited to the minimum size of the chip that can be measured and the dispersion or number of such chip points on the tool.

Studies that have adopted image sensing as the input data source for analysis of the tool condition have shown this is also a daunting approach due to the obstructive nature of the cutting region, which allows contamination, hence reducing the viability of the captured image to accurately and precisely depict the condition of the tool. Besides the mentioned drawbacks, the variability caused by the different conditions surrounding image/vision sensing, such as illumination, and variation in pixel senses of different devices, have also limited the approach in the propagation of signals that can intelligently depict the tool and workpiece condition.

The approach in this study focused on advanced signal processing and analysis, and feature extraction, selection, and classification, to determine the condition of the cutting tool during the machining operation without intermittent stoppage of machining. In this study, the cutting tool was first classified into four (4) classes using ISO 3685:1993 based on the flank wear (VB) of the cutting tool. The captured vibration signals were first decomposed using the EMD method to obtain the corresponding IMFs and residuals of the signals. To derive the feature vectors of the signals with instantaneous properties capable of classifying the tool classes, the HHT function was applied to the IMFs. It was observed that 12 features were obtained after applying the Hilbert transform to the IMFs. To reduce the computational burden and time, feature selection was performed with GA. After applying three (3) ML models to classify the tool conditions, with 5-fold and 10-fold cross-validation, the following observations were noted:

1. Neural network feed-forward backprop with an SCG ML model was first adopted to classify the tool classes, with a fair error of 0.102. A better model was observed to provide better performance.
2. SVM and KNN were applied to feature classification with both 5-fold and 10-fold cross-validation, and the effectiveness of the models was evaluated by determining the error loss of both models.
3. The lowest error loss of 0.4752 was observed with the SVM model when 5-fold cross-validation was implemented, whereas, with the KNN model, the lowest error loss was observed to be 0.0166 when 5-fold cross-validation was implemented. In this case, feature selection using GA was implemented before ML classification.
4. The lowest error loss of 0.4881 was observed with the SVM model when 10-fold cross-validation was implemented, whereas, with the KNN model, the lowest error loss was observed to be 0.0109 when 10-fold cross-validation was implemented. Similarly, feature selection using GA was implemented before ML classification.
5. When all the features were used (no feature selection was performed), the lowest error loss of 0.1170 was observed for the SVM model when 10-fold cross-validation was implemented, whereas, with the KNN model, the lowest error loss was observed to be 0.1606 when 10-fold cross-validation was implemented.
6. Moreover, when all the features were used (no feature selection was performed), the lowest error loss of 0.1021 was observed for the SVM model when 5-fold cross-validation was implemented, whereas, with the KNN model, the lowest error loss was observed to be 0.1870 when 5-fold cross-validation was implemented.

7. Of the two models, KNN performed better in classifying the tool classes during the machining operation when the decomposition method was applied to the vibration signals captured during the operation.
8. SVM models performed better when all the features extracted from vibration signals were considered, compared to when feature selection was implemented, whereas, for the KNN model, the performance was better when feature selection was implemented.
9. The methodology developed based on tool classification using advanced signal processing techniques can be used to classify product quality output based on work requirements in terms of roughness parameters.

The authors expect that, if varying vibration signals can be captured with different classes of tool conditions (based on quality requirement) and their corresponding workpiece roughness parameters, the KNN ML model can be implemented to monitor the tool condition and the corresponding product quality. The limitation of this study, however, is that it did not incorporate machine tool malfunctioning in the signal analysis. It will be useful to accommodate this factor as it may impact the product output and the vibration signals captured during machining.

**Supplementary Materials:** The following supporting information can be downloaded at: https://www.mdpi.com/article/10.3390/app13042248/s1, figures, tables and data, can be downloaded within the Supplementary Material. Table S1: DataMonitoring(vib & Time); Table S2: ErrorPrediction; Table S3: HilbertFeatures.

**Author Contributions:** I.O.O.: Conceptualization, writing of draft preparation; O.A.O.: Review and editing. All authors have read and agreed to the published version of the manuscript.

**Funding:** This research received no external funding.

**Data Availability Statement:** Data is contained within the Supplementary Material.

**Acknowledgments:** This is to acknowledge the support from the lab technicians during the experimentation and also the support staff of Smart2Devices, Ameer who provided IoT gateways and sensor connections used in the experiments.

**Conflicts of Interest:** The funders had no role in the design of the study; in the collection, analyses, or interpretation of data; in the writing of the manuscript; or in the decision to publish the results.

## Abbreviations

| | |
|---|---|
| TCM | Tool Condition Monitoring |
| FFT | Fast Fourier Transform |
| DWT | Discrete Wavelet Transform |
| SVM | Support Vector Machine |
| KNN | K-Nearest Neighbor |
| SCG | Scaled Conjugate Gradient |
| ML | Machine Learning |
| HHT | Hilbert–Huang Transform |
| TWCM | Tool and Workpiece Condition Monitoring |
| WT | Wavelet Transform |
| RW | Roulette Wheel |
| VMB | Variational Mode Decomposition |
| LSVM | Linear Support Vector Machine |
| PCA | Principal Component Analysis |
| LR | Logistic Regression |
| ANN | Artificial Neural Networks |
| CNN | Convolutional Neural Network |

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
