# Peer review of "Tool and Workpiece Condition Classification Using Empirical Mode Decomposition (EMD) with Hilbert–Huang Transform (HHT) of Vibration Signals and Machine Learning Models"

_applsci, doi:10.3390/app13042248_

Round 1
Reviewer 1 Report
The paper aims to apply some advanced signal processing techniques on the vibration signals captured experimentally during machining operation for the decision making and analysis. However, the novelty and scientific soundness is very poor. The quality of presentation is not good enough for publication.
First of all, the authors should spend some time checking and revising the format of paper draft. For example, some unnecessary or unrelated texts should be deleted before submission.
Secondly, the image quality is very poor. To be more specific, the resolution of figures in this paper needs to be further enhanced. Besides, in Figure 2, the red lines are blocked by the legend, which is inappropriate.
In addition, the authors fail to introduce their methods adequately and the relevant results are not clearly presented. Therefore, there are many things requiring further check and revision before it can be considered for possible publication.
Reviewer 2 Report
Manuscript Number: applsci-2158506
Title: Evaluating the Tool and Workpiece Condition Via Vibration Signal Processing Using Machine Learning
Decision: Minor revision
Article Type: Article
The article is, in general, well written but there are some issues that article should consider to revise in order to improve its quality. Some comments were done in this way:
Ø Keywords should be edited
Ø The chemical composition of the workpiece material should be given.
Ø Cutting tool form and coating properties are important parameters that affect vibration, wear, cutting temperature and chip breaking. All specifications of the cutting tool should be given.
Ø Fig 1. The image quality is poor. Also give more details.
Ø The exact location of the vibration sensor should be specified. The cutting tool should be given with the feed and the direction of rotation of the workpiece.
Ø Cutting tool wear has a significant effect on vibration. Used tool wear should be measured. Also, how many different used tools were tested for the experiments? What were the wear differences of these suits? These directly affect the vibration. Authors should clarify this situation.
Ø All the features of the vibration sensor should be given.
Ø Fig. What do the colors mean in 2? It should be declared on the figure.
Ø Fig 8. The image quality is poor.
Ø What is the program used for Machine Learning? How many variables and data were used?
A paragraph should be opened and explained in detail. After making the above corrections would recommend this article for publication in Applied Sciences.
Reviewer 3 Report
Dear author(s), please find below suggestions that may justify my final evaluation of the reviewed manuscript ‘Evaluating the Tool and Workpiece Condition Via Vibration Signal Processing Using Machine Learning, Manuscript ID: applsci-2158506.
Generally, the paper idea is interesting, the topic is up-to-date and, but needs improvement to be accepted in reputed journals.
1. Define the abbreviation first and use it in the text see line 18.
2. Insert a reference in line 36.
3. The literature review part needs the inclusion of more papers published on tool wear. A few examples are: https://doi.org/10.1016/j.triboint.2022.107773, https://doi.org/10.1007/s00170-022-09037-y, 10.24200/SCI.2016.3927, https://doi.org/10.1016/j.triboint.2022.107497, https://doi.org/10.3390/su14127472 and many more papers.
4. Resolution of figures is poor need high and good quality figures.
5. In Figure 2, adjust the colors as per the tool used in the study.
6. The figures' text does not match the paper's body text.
7. How the authors classified the four types of tools based on the wear level and types of wear?
8. Would be better for the reader to include the SEM or LM images of the four types of tools used in the study.
9. Results and discussion section lacks depth analysis,
10. Conclusions need to be more conclusive and also mentioned the limitations of the work.
Reviewer 4 Report
After reading submitted manuscript, my comments are as follows :
1. Title of manuscript is very general. It is not reflecting the algorithms used in paper. It should be carefully reframed.
2. Research objective should be modified. Extracting features from vibration signal can't be research objective. Rather, it is a procedure. Authors should refer to the standard published paper.
3. Introduction and literature review should be a single section rather than a separate section.
4. There should be a methodology chart which reflects the overall procedure of manuscript.
5. Instead of applying filter, authors should explore signal processing methods like Wavelet, EMD etc. Further, the extracted features should be listed in a separate table.
6. Authors should apply feature selection technique so that relevant features should be identified for prediction. It is computationally expensive to use 22 and 17 features. There is a strong possibility that with reduced feature set, classification accuracy may be enhanced. Kindly refer following journal and add in revised manuscript :
a. https://www.mdpi.com/2075-1702/10/3/176
7. It is always a best practice to use whole dataset for training and same dataset for 10-fold cross validation so that unbiased and reliable prediction results can be reported. Kindly refer 6a to get more idea.
8. All figures quality need to be improved for better readability.
9. Manuscript should be prepared as per format of journal. There are so many changes required in manuscript.
10.There seems to be no utility of screenshot of Excel file.If it is needed kindly address it.
Round 2
Reviewer 1 Report
The authors have successfully revised the manuscript by addressing comments.
Reviewer 3 Report
Dear Authors, Thanks a lot for including all the suggestions. I will accept it in its current form.
Reviewer 4 Report
Authors have incorporated suggestions given by reviewer and accordingly revised whole manuscript.